# Infestation of Oriental House Rat (*Rattus tanezumi*) with Chigger Mites Varies along Environmental Gradients across Five Provincial Regions of Southwest China

**DOI:** 10.3390/ijerph20032203

**Published:** 2023-01-26

**Authors:** Yan-Ling Chen, Xian-Guo Guo, Fan Ding, Yan Lv, Peng-Wu Yin, Wen-Yu Song, Cheng-Fu Zhao, Zhi-Wei Zhang, Rong Fan, Pei-Ying Peng, Bei Li, Ting Chen, Dao-Chao Jin

**Affiliations:** 1Institute of Pathogens and Vectors, Yunnan Provincial Key Laboratory for Zoonosis Control and Prevention, Dali University, Dali 671000, China; 2Institute of Microbiology, Qujing Medical College, Qujing 655000, China; 3Institute of Entomology, Guizhou University, Guiyang, Guizhou 550025, China

**Keywords:** ectoparasite, rodent, chigger mite, *Rattus tanezumi*, infestation, environmental heterogeneity

## Abstract

Chigger mites are the exclusive vector of scrub typhus. Based on field investigations of 91 survey sites in 5 provincial regions of Southwest China, this paper reported variations of chigger infestation on the oriental house rat (*Rattus tanezumi*) along various environmental gradients. A total of 149 chigger species were identified from 2919 *R. tanezumi* in the 5 provincial regions, and *Leptotrombidium deliense* (a major vector of scrub typhus in China) was the first dominant chigger species, followed by *Ascoschoengastia indica* and *Walchia ewingi*. *Rattus tanezumi* had a stable overall prevalence (*PM* = 21.10%), mean abundance (*MA* = 7.01), and mean intensity (*MI* = 33.20) of chiggers with the same dominant mites in the whole Southwest China in comparison with a previous report in Yunnan Province, but chigger infestations on *R. tanezumi* varied along different environmental gradients. *Rattus tanezumi* in mountainous landscape had a higher infestation load of chiggers with higher species diversity than in flatland landscape. The infestation was higher at lower altitudes and latitudes. A high intensity of vegetation coverage was associated with high infestations. The results reflect the environmental heterogeneity of chiggers on the same host species. Warm climate and high relative humidity are beneficial to chigger infestation on *R. tanezumi*.

## 1. Introduction

Chigger mites (Trombidiformes: Trombiculidae) are a group of tiny arthropods with over 3000 species recorded worldwide [1]. In the life cycle of chigger mites, only the larvae (often called chiggers) are ectoparasites of other animals, especially vertebrates (hosts) [2,3,4]. Chiggers are the exclusive vector of *Orentia tsutsugamushi*, the causative agent of scrub typhus (tsutsugamushi disease) [5]. Scrub typhus is an ancient vector-borne disease and zoonosis (zoonotic disease) widely distributed in China, Southeast Asia, and the Asia-Pacific region, threatening about one billion people worldwide [5,6,7].

Rodents and other small mammals (e.g., shrews and tree shrews) are the most common hosts of chiggers [4,8]. The oriental house rat or Asian house rat, *Rattus tanezumi* (Temminck, 1844), is a common species of commensal rodents that are often found in residential areas and agricultural areas (farmlands and other types of cultivated lands) and nearby environments [9,10,11]. Besides destroying crops and other agricultural plants as an important agricultural pest, *R. tanezumi* is also closely associated with zoonoses such as plague, murine typhus, scrub typhus, hemorrhagic fever with renal syndrome (HFRS), rat-bite fever (RBF), leptospirosis, and other zoonoses [10,12,13,14,15]. As a common rodent host of ectoparasites, *R. tanezumi* often harbors lots of fleas and chiggers on its body surface (skin), and some vector-borne diseases (e.g., plague, murine typhus, and scrub typhus) can be transmitted from the rat to humans through the biting activity of fleas and chiggers [16,17]. Therefore, it is of medical importance to study chiggers and some other ectoparasites on *R. tanezumi*.

A previous study in our research group once reported the infestation and related ecology of chigger mites on *R. tanezumi* in Yunnan Province, which is one of five provincial regions of Southwest China. The original data of the previous study came from the field investigation in only one province (Yunnan) between 2001 and 2015, not covering all the 5 provincial regions of Southwest China [18]. To expand and deepen the previous study, the field investigations of the present study between 2001 and 2019 have covered all the 5 provincial regions of Southwest China and the survey sites have increased from the previous 34 sites to the present 91 sites. The present study is an attempt to answer the following questions: (1) Will the overall infestation of chigger mites on the same host species (*R. tanezumi*) fluctuate significantly when sampling sites and survey scope are largely expanded? (2) Will the dominant species of chigger mites on *R. tanezumi* vary greatly without a relative stability? (3) How do infestation indices and community parameters of chigger mites on *R. tanezumi* vary with different environments? Besides answering the above questions, the present study will update and expand the knowledge of chigger infestation on *R. tanezumi* and further provide more detail and comprehensive information for the surveillance of chigger mites and some other related studies in Southwest China.

## 2. Materials and Methods

### 2.1. Field Investigations and Sampling of Chigger Mites

Between 2001 and 2019, field investigations were conducted in 91 survey sites across 5 provincial regions of Southwest China, including Yunnan, Guizhou, Sichuan, Tibet (Xizang), and Chongqing. The 91 survey sites covered different latitudes, altitudes, landscapes, and habitats. The related climatic parameters at each survey site were obtained from the National Earth System Science Data Center, National Science & Technology Infrastructure of China (http://www.geodata.cn), which included monthly mean temperature, precipitation, relative humidity, and normalized different vegetation index (NDVI, an indicator of the intensity of vegetation coverage at a given location) for the survey month. Mousetraps (Guixi Mousetrap Apparatus Factory, Guixi, Jiangxi, China) were placed at each survey site to sample animal hosts (rodents and other small mammals) in the late afternoon and checked the next morning. Each sampled animal host was separately placed in a white cloth bag and then transferred to the field laboratory and was anaesthetized with ether (cotton balls soaked with ether) within a closed container. Most rodent pests sampled from agriculture and forestry areas were euthanized because the local government encourages their eradication. Some non-pest small mammals (e.g., weasels, moles, and some squirrels) were anaesthetized for two to five minutes, according to their body size, and they were released into the wild after processing. Under anesthesia, chiggers were conventionally collected from each animal host by using a curette or lancet on a large white tray. All collected chiggers from each host were fixed in 70% ethanol. To avoid cross contamination, the white tray and collecting tools were cleaned and disinfected after each collection. Each animal host was identified into species according to its morphological characteristics and some body measurements [9]. In the laboratory, collected chiggers were mounted on glass slides with Hoyer’s medium after dehydration and transparency. Based on identification keys and the taxonomic literature [8,19], slide-mounted chiggers were identified to species level under an Olympus CX31 trinocular microscope (Olympus Corporation, Tokyo, Japan). After the identification of chiggers and their small mammal hosts, *R. tanezumi* and chiggers on its body surface were chosen as the target of the present study. The sampling and research of animals involved were officially approved by the local wildlife affairs authority and Animal Ethics Committee of Dali University (Ethics approval number DLDXLL2020-1104). Representative specimens of animal hosts and chigger mites are deposited in the specimen repository of the Institute of Pathogens and Vectors of Dali University.

### 2.2. Statistical Analysis

The constituent ratio (*C_r_*), prevalence (*PM*), mean abundance (*MA*), and mean intensity (*MI*) as indicated in the equations below were used to calculate the infestation of *R. tanezumi* with chiggers [20,21]. Differences in chigger prevalence (*PM*) on *R. tanezumi* in different environments were analyzed by using chi-square test. Mann–Whitney U test and Kruskal–Wallis test were adopted in the analysis of differences in mean abundance (*MA*) and mean intensity (*MI*) of chiggers in different environments. The above statistical analyses were performed under the statistical software SPSS (v.25). *p* values < 0.05 were considered statistically significant. Species richness and diversity of chigger community on *R. tanezumi* were computed using three community indices, namely the richness index (*S*), Margalef index (*R*), and Shannon–Wiener diversity index (*H’*) [18,22], as indicated in the equations below.
(1)$$C_{r}=\frac{N_{i}}{N}\times 100\%$$
(2)$$PM=\frac{H_{i}}{H}\times 100\%$$
(3)$$MA=\frac{N_{i}}{H}$$
(4)$$MI=\frac{N_{i}}{H_{i}}$$
(5)$$S=\sum S_{i}$$
(6)$$R=\frac{\left(S-1\right)}{In\;N}$$
(7)$$H’=-\sum_{i=1}^{S}\left(\frac{N_{i}}{N}\right)ln\left(\frac{N_{i}}{N}\right)$$

In all the above formulas, *N_i_* = number of a certain chigger species (species *i*) on a certain species of host (*R. tanezumi* in the present study), *N* = total number of chigger individuals, *H* = total number of hosts captured, *H_i_* = number of hosts infested with chigger mites, and *S_i_* = a certain chigger species *i* within the chigger community on *R. tanezumi* [18,20,21,22].

## 3. Results

### 3.1. Sampling and Identification of Chigger Mites on R. tanezumi

A total of 2919 oriental house rats, *R. tanezumi*, were captured from 56 (56/91) of the 91 survey sites sampled across 5 provincial regions of Southwest China, including Yunnan, Guizhou, Sichuan, Tibet (Xizang), and Chongqing (Figure 1, Table A1). Of the 20,604 chiggers collected from the body surface of *R. tanezumi*, 20,453 were identified into 149 species (*S* = 149) and 19 genera in 2 families (Trombiculidae and Leeuwenhoekiidae). The remaining 151 chiggers were unidentified because of the absence of key characters (broken body), key characters not clear due to debris, or suspected new species. The unidentified 151 chiggers were not included in statistical analyses in the present study. Here, chiggers are the larvae of chigger mites, which were collected from the body surface of *R. tanezumi*. The scientific names and collected individuals of 149 chigger species are listed in Appendix Table A2.

### 3.2. Overall Infestation and Dominant Species of Chiggers on R. tanezumi

Of the 2919 rats (*R. tanezumi*), 616 were infested with chiggers, and overall infestation indices (overall prevalence *PM*, mean abundance *MA*, and mean intensity *MI*) of all identified 149 chigger species on *R. tanezumi* were *PM* = 21.10% (616/2919), *MA* = 7.01 chiggers/rat (20,453/2919), and *MI* = 33.20 chiggers/rat (20,453/616) (Table 1). Of all 149 identified chigger species, *Leptotrombidium deliense*, *Ascoschoengastia indica*, and *Walchia ewingi* were three dominant mite species (Figure 2, Table 1). The total constituent ratio of three dominant chigger species reached 61.34% (*C_r_* = 61.34%) of the total 149 species. The *PM* and *MA* of *L. deliense* were higher than those of *A. indica* and *W. ewingi* (*p* < 0.001). The *MI* of *A. indica*, however, was higher than that of *L. deliense* and *W. ewingi* (*p* < 0.05) (Table 1).

### 3.3. Variation in Chigger Infestation along Environmental Gradients

The *PM*, *MA*, and community parameters (*S*, *R*, and *H’*) of chiggers on *R. tanezumi* from the mountainous landscape were much higher than those in the flatland landscape (*p* < 0.001). The *MI* of chiggers in the flatland landscape, however, was higher than that in the mountainous landscape (*p* < 0.001) (Table 2).

All infestation indices of chiggers on *R. tanezumi* at altitude ≤ 500 m were higher than those at other altitudes (*p* < 0.001). The community parameters at altitudes between 1501 and 3350 m were higher than those at other altitudes (Table 2).

The individuals of chiggers from *R. tanezumi* at latitude < 24° N accounted for 67.28% (*C_r_* = 67.28%) of total chiggers, which was much higher than that at other latitudes. The *PM* and *MA* of chiggers at latitude < 24°N were higher than those at other latitudes (*p* < 0.001). The *MI* at latitude > 26° N was higher than that at other latitudes (*p* < 0.05). The *H’* at latitude > 26° N was also higher than that at other latitudes. The *S* and *R* at latitudes between 24 and 26° N were higher than those at other latitudes (Table 2).

A total of 1093 *R. tanezumi* were sampled at longitudes between 100 and 102° E with a high constituent ratio (*C_r_* = 67.31%), and 317 rats were infested with 13,766 chiggers. All the infestation indices (*PM*, *MA*, and *MI*) and community parameters (*S*, *R*, and *H’*) of chiggers at longitudes between 100 and 102° E were higher than those at other longitudes (*p* < 0.001) (Table 2).

The infestation of *R. tanezumi* with chiggers also varied significantly at different climatic zones. The *PM* at between 50 and 100 mm precipitation zones was higher than that at other precipitation zones (*p* < 0.001). The *C_r_*, *MA*, and *MI* at climatic zones with between 100 and 200 mm of precipitation were higher than those at other precipitation zones (*p* < 0.05). The community parameters at <50 mm precipitation zone were higher than those at other precipitation zones (Table 2).

*Rattus tanezumi* harbored more chiggers (11,402 individuals) at between 20 and 25 °C temperature zones than at other temperature zones. The *PM* and *MA* of chiggers at between 20 and 25 °C temperature zones were much higher than those at other temperature zones (*p* < 0.001). The *MI* at between 20 and 25 °C temperature zones was also higher than that at other temperature zones, but it was not statistically significant (*p* > 0.05). The community parameters at <20 °C temperature zones were higher than those at other temperature zones (Table 2).

The *MI* at the humidity zone with between 60 and 70% of relative humidity was higher than that at other humidity zones (*p* < 0.001). The *PM* and *MA* of chiggers at humidity zones with ≥80% relative humidity were higher than those at other humidity zones (*p* < 0.001). The community parameters at the humidity zone with between 70 and 80% relative humidity were higher than those at other humidity zones (Table 2).

The infestation of *R. tanezumi* with chiggers was significantly affected by normalized different vegetation index (NDVI). There were more chiggers with higher infestation indices on *R. tanezumi* when NDVI was between 4000 and 5000 in comparison with those at other NDVI values (*p* < 0.001). Community parameters at NDVI ≤ 4000 were higher than those at other NDVI values (Table 2).

## 4. Discussion

### 4.1. Overall Infestation of Chiggers on R. tanezumi in Southwest China

Chiggers are the only ectoparasitic stage in the life cycle of chigger mites [2,3,4], and they are the target stage of the present study. As a typical commensal rodent species, *R. tanezumi* is widely distributed in China and some East and Southeast Asia countries, and it is one of the dominant rodent species in Southwest China [9,18]. Southwest China is a special geographical region with complex topography and climate, and it covers 5 provincial regions, namely Sichuan, Chongqing, Guizhou, Yunnan, and Tibet (Xizang) [23,24]. Based on field investigations of 91 survey sites between 2001 and 2019 in 5 provincial regions of Southwest China, the present study reported the species diversity (total number of species) and overall infestation of chiggers on *R. tanezumi* in the region. The overall infestation indices (*PM* = 21.10%, *MA* = 7.01, and *MI* = 33.20) of chiggers on *R. tanezumi* in 5 provincial regions of Southwest China in the present study were very close to those (*PM* = 20.9%, *MA* = 6.2, and *MI* = 29.8) in only one province (Yunnan) in the previous study [18]. These results imply that the overall infestation of *R. tanezumi* with chiggers seems stable in the whole Southwest China without a prominent fluctuation. The overall infestation of chiggers on the same host species (*R. tanezumi*) seems to not fluctuate significantly when the sampling sites and survey scope are largely expanded. A total of 149 species (*S* = 149) and 19 genera in 2 families were identified from *R. tanezumi* in 5 provincial regions of Southwest China in the present study, which are 18 species more than the number of chigger species (131 species and 19 genera in 2 families) in Yunnan Province alone in the previous study [18]. Although the scope of field investigation was largely expanded from only 1 province (Yunnan) with 34 survey sites in the previous study to 5 provincial regions with 91 survey sites in the present study, the increased number of chigger species (18 species) was limited, neither doubled nor tripled [18]. The results suggest that the majority of common chigger species may have been sampled in the present study and some rare chigger species may be too rare to be found in sampled field investigations.

Compared with other rodent species in the same region, *R. tanezumi* harbored more individuals and species of chiggers with a high rate of infestation. In the present study, the species richness (*S*) of chiggers on *R. tanezumi* was higher than that on the Chevrier’s field mouse, *Apodemus chevrieri* (*S* = 107 species), in Southwest China [25]. Some infestation indices of chiggers on *R. tanezumi* in the present study (*MA* = 7.01 and *MI* = 33.20) were also higher than those on *A. chevrieri* (*MA* = 6.32 and *MI* = 19.77) in the same region (Southwest China) between 2001 and 2019 [25]. The species richness and infestation indices of chigger mites on *R. tanezumi* in the present study were much higher than those on the striped field mouse, *Apodemus agrarius* (*S* = 14 species, *PM* = 3.4%, *MA* = 0.36, and *MI* = 10.63), in Southwest China [26]. The results suggest that *R. tanezumi* is highly susceptible to chiggers, and it has a high potential to harbor many chigger species with a great burden of infestation. The species diversity and infestation of chiggers on rodents and other small mammal hosts are very different on different host species because of various biological characteristics of hosts [20,25,26]. The unique biological characteristics of *R. tanezumi* should be one of the factors related to the high species diversity and heavy infestations of chigger mites on the rat.

### 4.2. Dominant Chigger Species on R. tanezumi in Southwest China

*Leptotrombidium deliense*, *A. indica*, and *W. ewingi* were three dominant chigger species on *R. tanezumi* in 5 provincial regions of Southwest China, and *L. deliense* was the first dominant one with the highest *C_r_* (Figure 2; Table 1). In the previous study from only one province (Yunnan), the dominant chigger species on *R. tanezumi* were also *L. deliense* (*C_r_* = 27.2%, *PM* = 6.3%, *MA* = 1.7, and *MI* = 26.8), *A. indica* (*C_r_* = 26.0%, *PM* = 4.6%, *MA* = 1.6, and *MI* = 35.6), and *W. ewingi* (*C_r_* = 8.6%, *PM* = 2.6%, *MA* = 0.5, and *MI* = 20.8) [18]. The dominant chigger species and their infestation indices on *R. tanezumi* from 5 provincial regions of Southwest China in the present study were also close to those from only one province (Yunnan) in the previous study [18]. The results suggest that the dominant chigger species and their infestation on *R. tanezumi* seem stable in the whole Southwest China without a prominent fluctuation, although the scope of field investigation was largely expanded in the present study. Of the three dominant chigger species in the present study, *L. deliense* is the most important vector of scrub typhus in China, and *A. indica* is suspected as a potential vector of the disease [27,28]. The occurrence of *L. deliense* and *A. indica* with high constituent ratios on the body surface of *R. tanezumi* would probably increase the potential risk of transmitting scrub typhus from rats to humans through the biting activity of chiggers [27,28].

### 4.3. Variations of Chigger Infestation on R. tanezumi along Different Environmental Gradients

As a common rodent species, *R. tanezumi* has a strong environmental adaptability and it is widely distributed in various environments within its distributed regions [9]. The results of the present study showed that the infestation of *R. tanezumi* with chiggers varied along different environmental gradients (Table 2). This variation reflects the instability and heterogeneity of chigger infestation on the same rodent host species under different environmental conditions. The results showed that chigger infestation on *R. tanezumi* varied in different landscapes. Although the sampled number of *R. tanezumi* in the flatland landscape was much more than that in the mountainous landscape, the infestation indices (*PM* and *MA*) and all the community parameters (*S*, *R*, and *H’*, the indices mainly reflecting the species richness and diversity of community) were higher in the mountainous landscape than in the flatland landscape, except *MI* (Table 2). The results of the present study are consistent with the previous study on the Chinese mole shrew (*Anourosorex squamipes*) in the same region, in which the species richness and infestation indices of chiggers on the shrew were higher in the mountainous landscape than in the flatland landscape [29]. Other previous investigations revealed that the rodent species in the mountainous landscape often have a higher infestation load of chiggers than in the flatland landscape [30,31]. A study on chigger distributions in Thailand showed that there was a much higher species richness in the mountainous uncultivated landscape than in the cultivated flatland landscape, which is similar to the result of the present study [32]. In Southwest China, the mountainous landscape is usually associated with complex topography and habitats, diverse vegetation, and high biodiversity without much human disturbance in comparison with the flatland landscape, and this may lead to the higher species diversity and infestation of chiggers on *R. tanezumi* in the mountainous landscape than in the flatland landscape [29,30,33].

The infestation indices (*PM*, *MA*, and *MI*) of chiggers on *R. tanezumi* at altitude ≤500 m were higher than those at other altitudes in 5 provincial regions of Southwest China (Table 2), and this is similar to the results in Yunnan Province [18]. The community parameters (*S*, *R*, and *H’*) which mainly reflect the species richness and diversity of chigger community, however, were higher at altitudes between 1501 and 3350 m than at other altitudes (Table 2). The higher infestation of chiggers on *R. tanezumi* at lower altitudes may be associated with *L. deliense* (the first dominant chigger species), which is mainly distributed at lower altitudes [28,34]. As a special geographical region, Southwest China has great different elevations and diverse climates. The temperature and humidity at lower altitudes are much higher than at higher altitudes, which is beneficial to the growth, development, and reproduction of *L. deliense* and other chigger species [28,34]. The different altitudinal gradients often have different topography, diverse climates, and various vegetation (e.g., forests and grasslands), and this may lead to differences in species richness of chiggers [35,36]. The species richness of chiggers in different altitudes in Southwest China may reflect the influence of environmental factors on chiggers, which may also be associated with different vegetation, temperature, humidity, and rainfall in different environmental gradients [25,37]. The host specificity of most chiggers is quite low, and some chigger species may cross-infest *R. tanezumi* from other animal species hosts in a complex environment [32,38,39].

The species diversity and infestation of chiggers on *R. tanezumi* varied along different latitudes, longitudes, and precipitation zones (Table 2). Most areas with lower latitudes in Southwest China have a hot and humid climate [28,30,40], which is beneficial to the growth, development, and reproduction of *L. deliense* and other chigger species [28,34], and this can partially explain the higher infestation (*PM* and *MA*) of *R. tanezumi* with chiggers at latitude <24° N in the present study (Table 2). The infestation (*MA* and *MI*) of *R. tanezumi* with chiggers at between 100 and 200 mm precipitation zones was higher than at other precipitation zones (Table 2), and this reflects the influence of precipitation on chiggers. Precipitation often influences the development, reproduction, and population dynamics of chiggers, and too much or too little precipitation is not ideal for most chigger species [32,41]. *Rattus tanezumi* harbored more chiggers with much higher infestation indices (*PM*, *MA*, and *MI*) at between 20 and 25 °C temperature zones than at other temperature zones (Table 2). The infestation indices at three humidity zones with >60% of relative humidity are obviously higher than those at the zone with ≤60% relative humidity. The *PM* and *MA* at humidity zones with ≥80% of relative humidity were significantly higher than those at other humidity zones (Table 2). These results reflect the influence of temperature and humidity on chigger infestations on *R. tanezumi*. The warm climate (20–25 °C) and high relative humidity (≥80%) should be an ideal environmental condition for chigger infestation on *R. tanezumi* in Southwest China. It is reported that hot and humid environments are optimal for the growth, development, and reproduction of most chigger mites including *L. deliense* [4,8,20,34], which is similar to the results of the present study.

The NDVI (normalized different vegetation index) used in the present study is a special index and it quantifies the intensity of vegetation coverage at a given location. It has previously been shown that most chiggers prefer humid environments with more than 80% relative humidity. Female chigger mites often lay their eggs on lower grass blades and leaves, and they spend most of their lives on vegetation that is no more than 30 cm above the ground level because of their needs for air humidity [42]. The results of the present study showed that infestation indices (*PM*, *MA*, and *MI*) of chiggers on *R. tanezumi* were significantly higher at between 4000 and 5000 NDVI values than at other NDVI values (Table 2), and this suggests that a relatively higher intensity of vegetation cover, which can provide chiggers with a well-vegetation covered and humid environment, is associated with the higher infestation of chiggers on *R. tanezumi* in Southwest China.

The results of the present study reveal the heterogeneity of chigger infestation on the same host species (*R. tanezumi*) in different environments such as different landscapes, latitudes, altitudes, vegetation coverages, and climatic zones. The temperature, precipitation, humidity, and vegetation often vary with elevation gradients and complex topographic conditions, and a complex interplay often happens among different environmental factors [36,43]. The variations of chiggers on *R. tanezumi* along different environmental gradients reflect the fact that chigger infestation on a certain rat species is usually affected by complex environmental factors, not a single one. From the present study, however, we are still unable to deduce a complex interplay of these environmental factors which influence chigger infestation, and more in-depth studies are still needed in future studies.

## 5. Conclusions

The oriental house rat (*R. tanezumi*) had a high potential to harbor many chigger species with high species diversity in 5 provincial regions of Southwest China, and it had a stable overall infestation of chiggers with the same dominant mite species (*Leptotrombidium delicense*, *Ascoschoengastia indica*, and *Walchia ewingi*) in comparison with a previous report in Yunnan Province. There is environmental heterogeneity in chigger infestations on *R. tanezumi*, with variations along various environmental gradients including different landscapes, latitudes, altitudes, vegetation coverages, climatic zones, etc. Warm and humid environments with high vegetation coverage are beneficial to chigger infestations on *R. tanezumi*.

## Figures and Tables

**Figure 1 ijerph-20-02203-f001:**
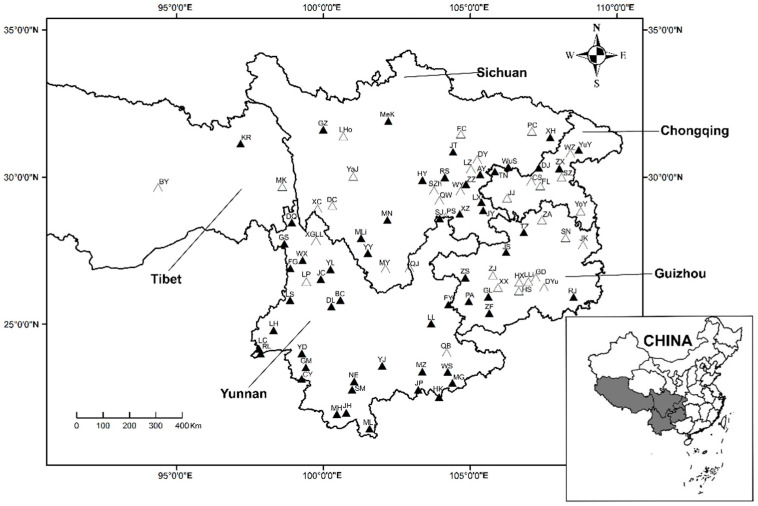
A total of 91 survey sites and sites where oriental house rats (*Rattus tanezumi*) were sampled in 5 provincial regions of Southwest China between 2001 and 2019 (symbols: filled black triangles represent sites where *R. tanezumi* were captured and hollow triangles represent sites without *R. tanezumi* captured).

**Figure 2 ijerph-20-02203-f002:**
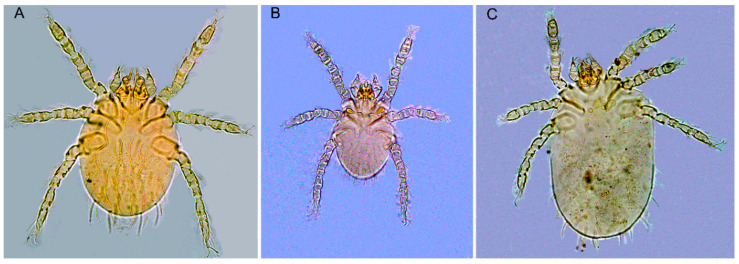
Photos of three dominant chigger species (10 × 40) on *Rattus tanezumi*. ((**A**) = *Leptotrombidium deliense*, (**B**) = *Ascoschoengastia indica*, and (**C**) = *Walchia ewingi*).

**Table 1 ijerph-20-02203-t001:** Infestations of three dominant chigger species on oriental house rats (*Rattus tanezumi*) sampled between 2001 and 2019 in 5 provincial regions of Southwest China.

Dominant Chigger Species	Constituent Ratios of Chiggers		Chigger Infestations on *Rattus tanezumi*		
	No.	*C_r_* (%)	*PM* (%)	*MA*	*MI*
*Leptotrombidium deliense*	6297	30.79	6.75	2.16	31.96
*Ascoschoengastia indica*	4483	21.92	4.32	1.54	35.58
*Walchia ewingi*	1766	8.63	2.77	0.61	21.80
All 149 identified chigger species	20,453	100.00	21.10	7.01	33.20

**Table 2 ijerph-20-02203-t002:** Variations of chigger infestation on oriental house rats (*Rattus tanezumi*) in different environments in 5 provincial regions of Southwest China (2001–2019).

Different Environments		Number of Examined *Rattus tanezumi*	Community Parameters of Chiggers			Constituent Ratios of Chiggers		Chigger Infestation on *Rattus tanezumi*		
			*S*	*R*	*H’*	No.	*C_r_* (%)	*PM* (%)	*MA*	*MI*
Landscapes	Mountainous	969	113	12.37	2.65	8566	41.88	28.69	8.84	30.81
	Flatland	1950	87	9.17	2.12	11,887	58.12	17.33	6.10	35.17
Altitudes	≤500 m	259	31	3.59	1.90	4301	21.03	34.36	16.61	48.33
	501–1000 m	1465	76	7.92	2.07	12,956	63.35	23.62	8.84	37.45
	1001–1500 m	575	58	8.33	2.82	939	4.59	13.83	1.63	12.04
	1501–3350 m	620	80	10.23	2.92	2257	11.04	16.61	3.64	21.91
Latitudes	<24° N	1223	89	9.23	2.22	13,761	67.28	32.54	11.25	34.58
	24–26° N	1461	90	10.42	1.96	5122	25.04	12.53	3.51	27.99
	>26° N	235	44	5.84	2.81	1570	7.68	14.89	6.68	44.86
Longitudes	<98° E	842	65	8.59	2.23	1719	8.40	15.56	2.04	13.12
	98–100° E	563	55	7.06	2.19	2097	10.25	17.76	3.72	20.97
	100–102° E	1093	87	9.02	2.26	13,766	67.31	29.00	12.59	43.43
	>102° E	421	38	4.65	1.84	2871	14.04	16.15	6.82	42.22
Precipitations	<50 mm	895	99	11.78	2.33	4090	20.00	16.54	4.57	27.64
	50–100 mm	515	75	8.74	2.22	4771	23.33	29.51	9.26	31.39
	100–200 mm	728	60	6.60	2.06	7655	37.43	21.98	10.52	47.84
	>200 mm	781	40	4.71	1.91	3937	19.25	19.97	5.04	25.24
Temperature	<20 °C	1120	115	13.53	2.85	4556	22.28	15.63	4.07	26.03
	20–25 °C	1091	74	7.81	2.21	11,402	55.34	27.50	10.45	38.01
	>25 °C	708	45	5.23	2.05	4495	21.82	19.92	6.35	31.88
Relative humidity	≤60%	153	16	3.85	1.96	49	0.28	7.19	0.32	4.45
	60–70%	1006	84	9.32	1.91	7395	36.16	17.30	7.35	42.50
	70–80%	1211	97	10.89	2.82	6735	32.93	22.13	5.56	25.13
	≥80%	549	39	4.35	1.65	6274	30.68	29.69	11.43	38.49
NDVI	≤4000	932	84	10.39	2.53	2936	14.35	13.95	3.15	22.58
	4000–5000	708	68	7.26	1.91	10,209	49.91	28.25	14.42	51.05
	5000–6000	1046	72	8.15	2.44	6063	29.64	21.32	5.80	27.19
	≥6000	233	36	4.91	2.42	1245	6.09	27.04	5.34	19.76

## Data Availability

The experimental data used to support the findings of this study are available from the corresponding author upon reasonable request.

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
