# Peer review of "Infestation of Oriental House Rat (Rattus tanezumi) with Chigger Mites Varies along Environmental Gradients across Five Provincial Regions of Southwest China"

_ijerph, 2023, doi:10.3390/ijerph20032203_

Round 1
Reviewer 1 Report
The manuscript describes a large body of work concerning the prevalence/incidence of chigger mites on rodents in China. The work reported was extensive and very detailed. My only comments would concern the Results section - I'm not sure how this can be streamlined/simplified, but there is a lot of detail there which is quite hard to follow.
While the standard of English is generally very good, there are a number of minor changes required. For example, the Tables use the full species name (rattus) despite these being used earlier.
Despite these details, I think the manuscript is suitable for publication in this journal.
Author Response
Many thanks to the reviewer for the positive comments and suggestions. According to the suggestions, we have made some revisions to the manuscript. To streamline/simplify the results section, we have deleted some redundant descriptions and merged the original Table 2 to Table 9 into a new Table 2. The revised results section has now been much shorter and more condensed than the former one. In addition, the original Figure 1 and Appendix 1 have been revised, and the revised Appendix 1 has become much simpler than the former one. Some minor errors in the text of the former manuscript have also been corrected. Please see the revised parts in red color in the revised manuscript.

Reviewer 2 Report
I was kindly invited to review the manuscript “Infestation of oriental house rat (Rattus tanezumi) with chigger mites varies along environmental gradients across five provincial regions of Southwest China” submitted for consideration to the International Journal of Environmental Research and Public Health.
This manuscript did a good job in analyzing variations of chigger infestation on the oriental house rat (Rattus tanezumi) along various environmental gradients based on field investigations in five provincial regions of Southwest China, which provide more detail and comprehensive information for the surveillance of chigger mites in Southwest China. However, I have some minor questions about the manuscript. First, I noticed that you emphasized “a vast geographical region” after “five provincial regions of Southwest China” in Line 60, which seems a little weird here. Cause I think compared to the world or China, five provinces in China are not a “vast geographical region”. Second, I suggested that you can merge Table 3 to Table 9 into one Table. Third, the variation of chigger infestation on R. tanezumi was usually affected by complex environmental factors, not a single factor. You’d better give consideration to the complex interplay of environmental conditions in the discussion. Finally, I think the “Geographical detector” method is more suitable for your study according to your study design, you can learn about it and consider using it in your future research.
Author Response
Response: Many thanks to the reviewer for the positive comments and suggestions. According to the suggestions, we have made some revisions to the manuscript. The inappropriate use of “a vast geographical region” has been revised. In combination of other reviewer’s suggestion, we have deleted some redundant descriptions and merged the original Table 2 to Table 9 into a new Table 2. The revised results section has now been much shorter and more condensed than the former one. At the same, some minor errors in the former manuscript have also been corrected. In the discussions section, we have added some descriptions about the complex interplay of environmental conditions. However, we are still unable to deduce the complex interplay of environmental factors which influence the chigger infestation from the results of the present study, and more in-depth researches remain to be done in the future studies. Please see the revised parts in red color in the revised manuscript.
